# Isolation, Structural Elucidation, In Vitro Anti-α-Glucosidase, Anti-β-Secretase, and In Silico Studies of Bioactive Compound Isolated from *Syzygium cumini* L.

**Adil Mujawah** [1,*], **Abdur Rauf** [2,*], **Sami Bawazeer** [3], **Abdul Wadood** [4], **Hassan A. Hemeg** [5] **and Saud Bawazeer** [6]

1 Department of Chemistry, College of Science and Arts, Qassim University, Ar Rass 51921, Saudi Arabia
2 Department of Chemistry, University of Swabi, Swabi, Anbar 23561, Khyber Pakhtunkhwa, Pakistan
3 Department of Pharmacognosy, Faculty of Pharmacy, Umm Al-Qura University, Makkah P.O. Box 42, Saudi Arabia
4 Department of Biochemistry, Abdul Wali Khan University, Mardan 23200, Khyber Pakhtunkhwa, Pakistan
5 Department of Medical Laboratory Technology, College of Applied Medical Sciences, Taibah University, P.O. Box 344, Al-Medinah Al-Monawara 41411, Saudi Arabia
6 Department of Pharmaceutical Chemistry, Faculty of Pharmacy, Umm Al-Qura University, Makkah P.O. Box 751, Saudi Arabia
* Correspondence: a.mujawah@qu.edu.sa (A.M.); abdurrauf@uoswabi.edu.pk (A.R.)

**Abstract:** Diabetes is one of the main health issues worldwide because of its lifetime duration. To overcome this health problem, the current study was conducted. This investigation aims to explore the α-glucosidase and β-secretase potential of extract/fractions and pure isolated compounds of *Syzygium cumini* bark. The chloroform extract of *Syzygium cumini* bark was subjected to chromatographic analysis to yield compound **1**. The structure of isolated phytochemical (**1**) was conducted using advanced spectroscopic analysis. Among test extracts, the chloroform fraction exhibited a significant effect against α-glucosidase with a % activity of 86.20% and an $IC_{50}$ of 77.09 μM, while the isolated compound exhibited a promising effect with a % activity of 91.54 and an $IC_{50}$ value of 17.54 μM. The extract/fractions and isolated compound **1** also showed promising effects against the β-secretase enzyme, having % effects of 83.21 and 91.54% with $IC_{50}$ values of 318.76 and 17.54 μM, respectively. The extract/fractions and compound **1** were found to possess promising inhibitory activity against α-glucosidase and β-secretase. This research project opens a new avenue for research into detailed chemical and biological studies on *Syzygium cumini* to isolate bioactive enzyme inhibitors. Furthermore, the isolated compound **1** friedelin was docked into the active site of β-secretase and α-glucosidase. The molecular docking was assessed using molecular docking via the MOE-Dock tool. The docking results showed good docking scores of −6.84 and −6.46 when docked against β-secretase and α-glucosidase, respectively, and strong interactions.

**Keywords:** *Syzygium cumini*; extracts; friedelin; α-glucosidase; *β-secretase*; in silico studies

## 1. Introduction

Medicinal plants have been used as traditional medicines for the treatment of various diseases around the world. Compounds derived from these plants have been extensively studied for their pharmacological properties and potential as novel drugs [1–3]. In recent years, there has been an increasing interest in investigating the therapeutic potential of plants found in different regions of the world [4,5]. Many wild plants have been shown to possess therapeutic properties and are widely used in traditional home remedies and as raw materials for the pharmaceutical industry [6,7]. *Syzygium cumini*, a member of the *Myrtaceae* family, is a tropical evergreen tree also known as black plum (Jaman). This plant is known for its fruits, woods, and decorative value. It is native to India and the neighboring regions of Southeast Asia, Sri Lanka, and Myanmar and can grow up to 30 m in height with a lifespan of over 100 years [8–12]. The unripe fruit juice of *S. cumini* is used

for the preparation of vinegar and is considered to be carminative, stomachic, and diuretic, while the ripe fruit is used for the creation of conservation, squashes, and jellies. The leaves of this plant have been used in traditional medicine as a remedy for diabetes mellitus in many countries [13–15]. The leaves are used to improve the health of teeth and gums and to cure fever, gastropathy, strangury, dermopathy leucorrhea, diabetes, and stomachalgia [16]. The fruit, bark, and leaves of this plant have been found to have anti-diabetic properties due to the presence of various phytochemicals such as flavonoids, phenolic acids, and anthocyanins. Studies have shown that *S. cumini* and its isolated compound friedelin have a hypoglycemic effect on individuals with type 2 diabetes. The plant has been found to reduce fasting blood sugar levels, improve insulin sensitivity, and lower oxidative stress in diabetic individuals [17]. It is thought that this is achieved through the inhibition of $\alpha$-glucosidase and $\alpha$-amylase, two enzymes involved in carbohydrate metabolism [18,19]. In addition, *S. cumini* and friedelin have also been found to have anti-Alzheimer effects. The plant has been found to reduce the accumulation of beta-amyloid protein in the brain, which is a hallmark of Alzheimer's disease. It is thought that the anti-Alzheimer effects are due to the antioxidant properties of the plant, which protect the brain from oxidative stress and neuronal damage. *S. cumini* and its isolated compound friedelin have a potential therapeutic effect on both diabetes and Alzheimer's disease. [1–3]. Friedelin is also documented as having hypolipidemic [4], analgesic, anti-inflammatory, antipyretic, and analgesic properties [5]. Friedelin is a pentacyclic triterpenoid, which is perhydropicene that is substituted by an oxo group at position 3 and methyl ($CH_3$) groups at positions 4, 4a, 6b, 8a, 11, 11, 12b, and 14a. It has previously been isolated from *Ageratum conyzoides*, *Azima tetracantha* Lam, and *Maytenus ilicifolia* [18–20]. It has been reported as an antipyretic, an anti-diarrheal, and a free radical scavenger and also possesses live protective effects [18,20]. Friedelin extracted from *Ageratum conyzoides* has been shown to reduce inflammation in rabbit eyes, while isolated from *Aucuba jabonica* possess excellent anti-inflammatory effect in various models such as histamine- and carrageenan-induced paw oedema models [18]. Friedelin is also previously documented for excellent antimicrobial activity [20].

This study aimed to isolate, characterize, and evaluate the bioactivity of compounds present in *S. cumini* concerning $\alpha$-glucosidase and $\beta$-secretase activities. This study involved the isolation of bioactive compounds from *S. cumini* L. using various chromatographic techniques. The structures of the isolated compounds were elucidated using spectroscopic and other analytical methods. In vitro, $\alpha$-glucosidase and $\beta$-secretase inhibitory activities of the compounds were evaluated. In silico studies were also carried out to explore the molecular interactions of the compounds with the target enzymes. The outcome of this study provided insight into the therapeutic potential of *Syzygium cumini* L and its active compounds for the management of diseases related to $\alpha$-glucosidase and $\beta$-secretase activities.

## 2. Results

### 2.1. Structure Elucidation of Compound *1*

Compound **1**, namely friedelin, was isolated from chloroform-soluble fractions of S. cumini bark. The chemical structure of the isolated compound (**1**; Figure 1) was elucidated by advanced spectroscopic techniques. The exact molecular formula of compound **1** was determined to be $C_{30}H_{50}O$ from the HREI-MS spectrum, which exhibited the [M]+ peak at m/z 426.3861 (calculated as MW 426.3911). The $^1$H-NMR and $^{13}$C-NMR spectra showed characteristic resonances for 30 carbons, including 8 $CH_3$, 11 $CH_2$, f4our CH, and 7 C. Similarly, $^1$H-NMR showed seven methyl groups appearing as a singlet and one methyl doublet, all in the up-field region ranges between $\delta$ 0.73 and 1.18 ppm. The methine protons appeared at C-4 as a quartet at $\delta$ 2.23 ppm, at C-8 as a doublet of doublets (dd) at $\delta$ 1.38 ppm, and at C-18 as a multiplet at $\delta$ 1.54 ppm. The broadband $^{13}$C-NMR spectra also showed the presence of carbonyl carbon (C=O) at $\delta$ 213.4 ppm. Detailed $^1$H and $^{13}$C-NMR data are presented in Table 1. The HMBC spectra also showed a correlation of methyl protons (H-23, $\delta$ 1.01 ppm) with C-5 ($\delta$ 38.0 ppm) and C-3 ($\delta$ 213.4 ppm), a correlation of other methyl pro-

tons (H-28, δ 1.18 ppm) with C-9 (δ 37.1 ppm) and C-8 (δ 53.2 ppm), while methyl protons (H-26, 27, δ 1.01, 1.05 ppm) showed correlations with C-13 (δ 38.3 ppm), 14 (δ 39.6 ppm) and 15 (δ 32.2 ppm). The methylene protons (H-2, δ 2.27/2.37 ppm) showed correlations with carbonyl carbon (C-3, C=O, δ 213.4 ppm), and methine carbon (C-10, δ 59.5 ppm). Detailed HMBC correlations are presented in Figure 2. The structure of friedelin (**1**) was identified by comparing the spectra data with previously reported data [21,22].

**Figure 1.** Chemical structure of friedelin **1** isolated from *Syzygium cumin* bark.

**Table 1.** $^1$H-NMR and $^{13}$C-NMR data of friedelin.

| Carbon No | $^1$H-NMR CHEMICAL Shift Values (δ in ppm)/Multiplicity | $^{13}$C-NMR Chemical Shift Values (δ in ppm) |
| --- | --- | --- |
| 1 | 1.95; 1.71/Overlapped | 18.8 |
| 2 | 2.37; 2.27/m | 36.2 |
| 3 | - | 213.4 |
| 4 | 2.23/q | 58.4 |
| 5 | - | 38.0 |
| 6 | 1.73; 1.26/Overlapped | 41.7 |
| 7 | 1.47; 1.38/m | 18.1 |
| 8 | 1.38/dd | 53.2 |
| 9 | - | 37.1 |
| 10 | 1.51/m | 59.5 |
| 11 | 1.43; 1.22/m | 35.5 |
| 12 | 1.31; 1.38/Overlapped | 31.9 |
| 13 | - | 38.3 |
| 14 | - | 39.6 |
| 15 | 1.45; 1.24/m | 32.2 |
| 16 | 1.53; 1.30/Overlapped | 35.5 |
| 17 | - | 30.0 |
| 18 | 1.54/m | 42.9 |
| 19 | 1.35; 1.21/m | 35.2 |
| 20 | - | 28.1 |
| 21 | 1.49/Overlapped | 32.2 |
| 22 | 1.49; 0.93/m | 41.4 |
| 23 | 1.01/d | 7.0 |
| 24 | 0.73/s | 14.2 |
| 25 | 0.87/s | 14.8 |
| 26 | 1.01/s | 18.1 |
| 27 | 1.05/s | 18.1 |
| 28 | 1.18/s | 32.6 |
| 29 | 0.95/s | 32.2 |
| 30 | 1.05 (s) | 32.2 |

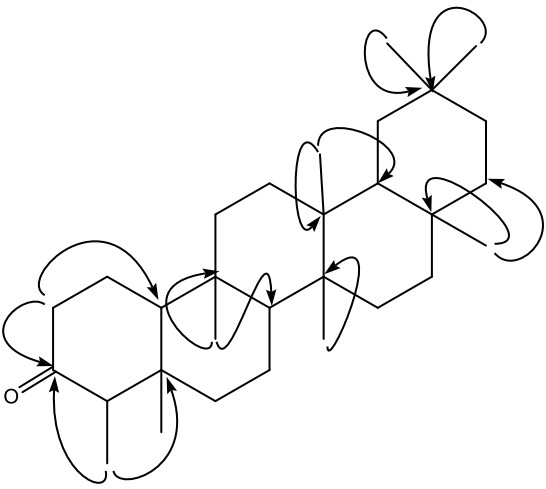

**Figure 2.** Key HMBC correlations of friedelin (**1**).

## 2.2. α-Glucosidase Inhibitory Effects

The α-glucosidase-inhibiting potential of crude extract and various fractions are given in Table 2. Among the tested extract/fractions the chloroform fraction exhibited promising effects with a % activity of 86.20% with an $IC_{50}$ of 77.09 μM, while friedelin exhibited a promising effect with a % activity of 91.54 and an $IC_{50}$ value of 17.54 μM.

**Table 2.** α-Glucosidase-inhibiting effects of crude extract/fractions and compound **1** extracted from *Syzygium cumini* bark.

| Samples/Standard | Concentrations (μg/mL) | % Inhibition | $IC_{50} \pm$ SEM (μM) |
|---|---|---|---|
| Hexane | 0.2 | 26.09 | NA |
| Chloroform | 0.2 | 86.20 | 77.09 ± 1.98 |
| Methanol | 0.2 | 79.32 | 87.54 ± 1.87 |
| Friedelin | 0.2 | 95.34 | 17.54 ± 1.54 |
| Standard | 0.2 | 91.34 | 23.07 ± 1.01 |

SEM stands for "standard error of the mean".

## 2.3. β-Secretase Inhibitory Effects

The results of the β-secretase-inhibiting activity are displayed in Table 3. The extract/fractions and isolated compoun1 also showed promising effects against the β-secretase enzyme having % effects of 83.21 and 91.54% with $IC_{50}$ values of 318.76 and 17.54 μM, respectively.

**Table 3.** β-Secretase-inhibiting effects of crude extract/fractions and compound **1** extracted from *Syzygium cumini* bark.

| Samples/Standard | Concentrations (μg/mL) | % Inhibition | $IC_{50} \pm$ SEM (μM) |
|---|---|---|---|
| Hexane | 0.2 | 48.12 | NA |
| Chloroform | 0.2 | 83.21 | 318.76 ± 2.09 |
| Methanol | 0.2 | 66.87 | 364.65 ± 1.73 |
| Friedelin | 0.2 | 91.54 | 290.43 ± 1.65 |
| Standard | 0.2 | 94.39 | 280.54 ± 1.43 |

SEM stands for "standard error of the mean".

## 2.4. Docking Results

The isolated compound was docked into the active site of β-secretase and α-glucosidase. The docking scores of the compounds were revealed as −6.84 and −6.46 when docked against β-secretase and α-glucosidase, respectively. The compound formed a total of two H-donor interactions with active site residue PRO 70 and one H–pi interaction with TRP 197 residue. The interaction was further compared to the reference compound berberine chloride, which indicates that the reference compound of β-secretase forms one hydrogen bond donor, one hydrogen bond acceptor, and one H–pi interaction with a docking

score of 6.621. The docking analysis of friedelin against α-glucosidase revealed that the compound formed a total of three H-donor interactions with the active site residues including ASP 231, ASP 233, and CYS 172. Furthermore, the reference compound acarbose was docked in the active site (glucosidase), where it formed three non-covalent interactions with the docking of −6.3214. Table 4 shows the docking score, interactions, energy, and distance of the compound against β-secretase and α-glucosidase. The 3D interactions of the compound within the active site of both receptors are present in Figure 3.

**Table 4.** Docking score and interactions of all the receptors with the isolated compound.

| Protein | Compound | Compound Atoms | | Interacting Residues | | Interaction Type | Distance | Energy (Kcal/mol) | Docking Score |
|---|---|---|---|---|---|---|---|---|---|
| α-glucosidase | friedelin **1** | C | 33 | ASP | 231 | H-donor | 3.75 | −0.1 | −6.4683 |
| | | C | 39 | ASP | 233 | H-donor | 3.72 | −0.1 | |
| | | C | 49 | CYS | 172 | H-donor | 3.80 | −0.2 | |
| α-glucosidase | acarbose | O | 18 | ASP | 408 | H-donor | 2.98 | −0.8 | −6.3214 |
| | | N | 19 | GLU | 304 | H-donor | 2.93 | −0.9 | |
| | | C | 39 | HIS | 279 | H-pi | 3.70 | −1.1 | |
| β-Secretase | friedelin **1** | C | 23 | PRO | 70 | H-donor | 3.75 | −0.2 | −6.8493 |
| | | C | 25 | PRO | 70 | H-donor | 3.72 | −0.4 | |
| | | C | 30 | TRP | 197 | H-pi | 3.80 | −0.2 | |
| β-Secretase | berberine chloride | C | 7 | GLY | 34 | H-donor | 3.67 | −0.1 | −6.621 |
| | | O | 3 | ILE | 118 | H-acceptor | 3.64 | −0.2 | |
| | | 6-ring | | SER | 35 | chloride | 3.48 | −0.3 | |

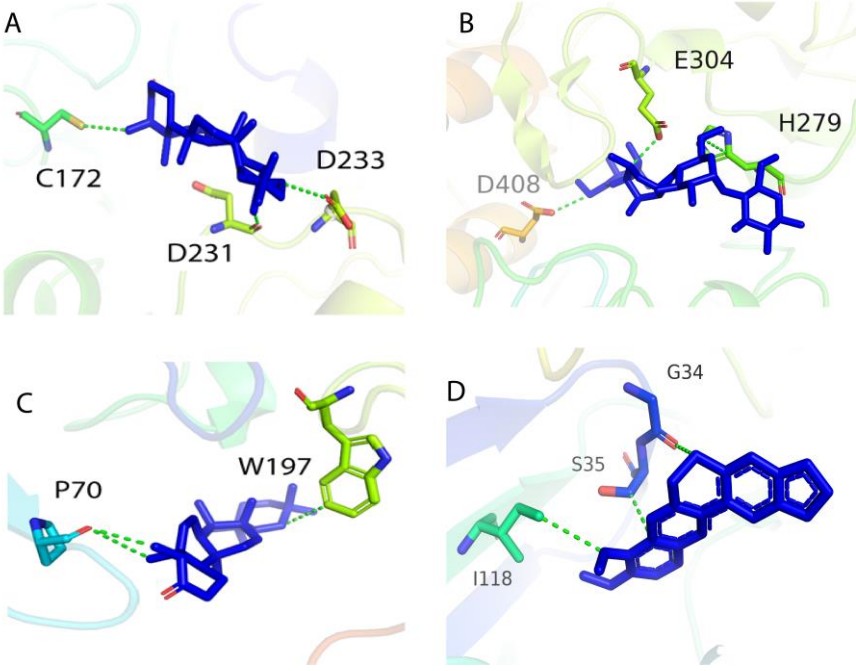

**Figure 3.** Three-dimensional interactions of compound **1** and reference compounds with the target protein: (**A**,**B**) α-glucosidase and (**C**,**D**) β-secretase.

### 2.5. ADMET Prediction

ADMET analysis has been carried out for the selected compound with effective IC$_{50}$ value and good docking score. The compound obeyed Lipinski's rules of five, according to which "a drug like compound must not have hydrogen bond acceptor more than 10, hydrogen bond donor not more than 5, water octanol coefficient not more than 10 and molecular weight must be less than 500 Daltons". It also has good ADMET properties in the required allotted limit. Table 5 shows the Lipinski rule of five and Table 6 shows the ADMET properties.

**Table 5.** Lipinski Rule of five.

| Compound ID | Hydrogen Bond Acceptor | Hydrogen Bond Donor | Log p | Molecular Weight (Da) |
|---|---|---|---|---|
| 1 | 1 | 0 | 8.457 | 426.729 |

**Table 6.** Pharmacokinetic (ADMET) properties.

| Category | Property with Unit | Compound 1 |
|---|---|---|
| Absorption | Water solubility (log mol/L) | −6.46 |
| | Caco2 permeability (log Papp in $10^{-6}$ cm/s) | 1.343 |
| | Intestinal absorption (%) | 100 |
| | Skin permeability (log Kp) | −2.578 |
| Distribution | VDss (human) (log L/kg) | 0.094 |
| | BBB permeability | 0.757 |
| | CNS permeability (log PS) | −1.657 |
| Metabolism | CYP2D6 substrate | No |
| | CYP3A4 substrate | Yes |
| | CYP1A2 inhibitor | No |
| | CYP2C19 inhibitor | No |
| | CYP2C9 inhibitor | No |
| Excretion | Total Clearance(log ml/min/kg) | −0.04 |
| | Renal OCT2 substrate | No |
| Toxicity | AMES toxicity | No |
| | Max. tolerated dose (human) (log mg/kg/day) | 0.546 |
| | Oral Rat Acute Toxicity (LD50) (mol/kg) | 2.387 |
| | Hepatotoxicity | No |
| | Skin Sensitization | No |
| | Minnow toxicity (log mM) | −3.15 |

## 3. Discussion

The chloroform-soluble fraction (22 g) of *Syzygium cumini* was subjected to repeated normal phase column chromatography using (hexane and EtOAc (85:15) as a mobile phase to yield compound **1** (Figure 4). The chemical structure of compound **1** was determined by using advanced spectroscopic analysis [22].

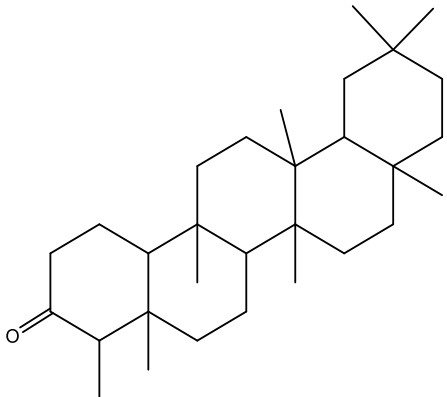

**Figure 4.** Structure of friedelin **1** isolated from *Syzygium cumini* bark.

Worldwide, the problem of diabetes mellitus is becoming alarming. Although a lot of antidiabetic molecules are in use for the treatment or management of diabetic mellitus, diabetes is still not controlled around the world. This failure of treatment is mostly associated with the side effects of available antidiabetic drugs which lead to poor compliance and the disease consequently entering its worst stage. To cope with these problems the search for new, effective, and safe antidiabetic molecules is a great task and challenge for medicinal chemists. The current research aims to find a good antihyperglycemic agent. Diabetes is accompanied by an increased blood glucose level and is considered a metabolic disorder. In this disorder, the beta cells of the pancreas produced deficient insulin [23]. In diabetes patients, several problems arise which comprise micro- and macrovascular

dysfunctions [24] or problems which curing using insulin and oral hypoglycemic agents and without side effects is very difficult [25]. α-Glucosidase inhibitors including carbose and miglitol, and miglitol is also involved in controlling postprandial hyperglycemia in diabetes mellitus patients [25]. To overcome the side effects and poor compliance with existing antidiabetics, several bioactive compounds derived from plants have been used for the treatment of diabetes alone or in combination with hypoglycemic agents [26]. Various parts of *Syzygium cumini* have been used in traditional systems for the curing of diabetes [27]. The main objective of this investigation was to validate the folk usage of *Syzygium cumini* for the treatment of diabetes and other disorder [28–30]. The results of the α-glucosidase-inhibiting effects indicated that in the number of tested extracts, the chloroform extract showed an inhibitory effect of 86.20% with an $IC_{50}$ value of 77.09, while the methanol extract showed excellent inhibitory activity at 79.32% with an $IC_{50}$ value of 87.54 μM. Among the tested samples, the isolated compounds showed a significant effect of 95.34% with an $IC_{50}$ of 17.54 μM in comparison to the standard drug, which has an effect of 91.34% and an $IC_{50}$ value of 23.07 μM. β-secretase, commonly known as the β-site AAP-cleaving enzyme, has been documented to be a transmembrane aspartic protease [31]. β-secretase is mainly involved in the pathogenesis of Alzheimer's disease (AD) [27]. BACE-1 is mainly responsible for the cleavage of AAP and the connected formation of β-amyloid, which is also associated with AD [32,33]. The discovery of a new and novel β-secretase inhibitor is an important therapeutic strategy for curing AD. Therefore, hexane, chloroform, and methanol extracts were screened for their β-secretase-inhibiting potential. Among the tested extracts, the chloroform extract showed an excellent effect of 83.21% with an $IC_{50}$ value of 318.76 μM, followed by the methanol extract of 66.87% with an $IC_{50}$ value of 364.65 μM. Compound **1** was found to be the most potent against BACE1, with a percentage inhibition of 91.54 and an $IC_{50}$ value of 17.54 μM. The standard drug showed a 94.39 % effect with an $IC_{50}$ value of 280.54 μM.

The inhibitory activity of friedelin was further validated by molecular docking analysis. The findings of the docking analysis reflect the strong inhibitory potential of friedelin by forming four hydrogen bond interactions in the active site residues. The ADMET prediction shows that this compound is not toxic and can be used as a drug molecule. The promising α-glucosidase- and β-secretase-inhibiting effects of compound **1** validate the traditional usage of *Syzygium cumini* for the treatment of diabetes and AD. In addition to the above aspect of this isolated compound, the ADMET study also showed good aspects. Conducting an ADMET study is very essential for the research of all molecules to save money as well as time. The journey of drug discovery is time-consuming as well as expensive. To save both time and money, using the best ADMET model is vital. In current study, a good GIT absorption was calculated from online software. A significant oral absorption makes a drug better with respect to the formulation of various dosage forms. Regarding distribution, the prediction of CNS and BBB distribution also makes a drug more attractive. The isolated compound was also found with no hepatotoxicity or skin toxicity, and AMES toxicity was also not predicted. However, further mechanistic studies are recommended.

## 4. Materials and Methods

### 4.1. Plant Collections

The bark of *Syzygium cumini* was obtained from the University of Swabi, KP, Pakistan. The specimen was identified by Dr. Muhammad Ilyas, Botany Department, Swabi University. The voucher specimen No. UOS/Bot103 of the collected plant was dropped in the herbarium of the Department of Botany Swabi University, Pakistan.

### 4.2. Extraction and Isolation

Ten kilograms of dried plant material was ground into a powder and subjected to cold extraction via maceration using 70% methanol as the solvent. The extraction was carried out at room temperature or slightly lower, but not heated, for a duration of two weeks. The resulting extract was then filtered and subjected to rotary evaporation under ambient

temperature and pressure, which yielded a crude extract weighing 107 g. The obtained extract was suspended in distilled water and treated with a polar and non-polar solvent which yielded hexane 32.98 g, chloroform 41.09 g, and methanolic extracts 44.98 g. Among the chloroform fractions, 22 g was assessed for column chromatography analysis using silica gel. The column was eluted with a mixture of two solvents (hexane and EtOAc; 85:15), which yielded a white needle-like crystalline compound **1**. The compound was purified by recrystallization in a mixture of hexane and acetone to obtain 99.94% pure compound **1** (Figure 4). The structure of the isolated compound was determined by using advanced spectroscopic analysis [22,34].

### 4.3. α-Glucosidase Inhibitory Screening

The α-glucosidase-inhibiting potential of extract/fractions and isolated compound **1** was performed according to the reported procedure [35]. A sonication of rat intestinal $(CH_3)_2$ acetone powder in saline having ration *w/v* (100:1) was conducted. Then, the supernatant was used as a wellspring of basic intestinal α-glucosidase after centrifugation. The solution for bioassay comprises 100 μL of each extract, fraction, or isolated compound (5 mg/mL) in dimethyl sulfoxide that was recreated in 100 μL of 100 mm phosphate buffer (pH = 6.8) in 96-well plates. The reaction mixture was hatched with intestinal α-glucosidase for five minutes with a 50 μL substrate (5 mM, p-nitrophenyl-a-D-glucopyranoside) as per the reported method. The substrate was mixed and the difference in absorbance at 400 nm was recorded by using a spectrophotometer. The screen extract/fractions and isolated compound **1** were substituted with DMSO (7.5%) in the control. The standard used in this assay was acarbose. The percentage effects were determined with the help of the below formula:

$$\% \text{ effect} = 100\text{-OD test well} \times 100/\text{OD control.}$$

### 4.4. β-Secretase Inhibitory Screening

The β-secretase-inhibiting effect of crude extract/fractions and the isolated compound was determined as per the reported method [30]. Buffer solution (82.5 μL) comprising sodium acetate buffer (50 mM) and having a pH of 4.50 was combined with every inhibitor in the wells, and then 2.5 μL of the stock solution of the enzyme was mixed in. The incubation of the reaction mixture was performed at 25 °C for 20 min. In the combined solution, the reaction will be started by adding 62.5 nM from the stock solution of the substrate. This solution was re-incubated at 37 °C for 60 min. Once the incubation of the 96-well plates was completed, 1000 nM of extracts/compounds solution from stock solution was added, and the incubation time of the plates was recorded with the help of a fluorometer machine according to the previously reported procedure [36]. The excitation and emissions for MCa are 325 and 400 nm, respectively.

### 4.5. Docking Methodology

The PDB structure of β-secretase receptors (PDB ID) was downloaded from the RCS PDB database and due to the non-availability of the 3D structure of α-glucosidase, the previously modeled structure by our lab group was used for the docking study. The target protein cannot be used for molecular docking directly because the structure might be coupled to heavy atoms, water, ligands, or cofactors. The water molecules were removed and polar charges were added to the structure [37]. The energy of both receptors was reduced using the default MOE energy minimization algorithm parameters (gradient: 0.05; force field: MMFF94X). The friedelin structures were built using the MOE-Builder module, and all structures were energy minimized. For the molecular docking studies, the default parameters of the MOE-Dock program were employed. The ligands were permitted to be flexible to determine the proper conformations of the ligands [38]. The best ligand conformations were examined for their binding interactions after docking using MOE and PyMol software [39].

*4.6. ADMET Prediction Methodology*

Ongoing drug discovery and design is a large economical risk if it is faced with some failures in different stages of development. These failures may be because of deficiencies in the drug's efficacy and safety, mainly related to its absorption, distribution, metabolism, excretion (ADME) properties and different toxicities (T). Therefore, the use of ADMET modelling is essential to lessen the chances of failure in drug discovery and development. pkCSM is an online server that easily performs six types of drug-likeness analyses including the Lipinski rule of five and a prediction model, 31 ADMET points analysis consisting of 3 basic properties, 6 absorptions, 3 distributions, 10 metabolisms, 2 excretions, and 7 toxicities. pkCSM provides free online server access at http://biosig.unimelb.edu.au/pkcsm/prediction.

*4.7. Statistical Analysis*

ANOVA software was used for statistical analysis. All data are presented as the mean ± SEM of three experimental results.

**5. Conclusions**

Phytochemicals derived from medicinal plants provide a promising molecular scaffold for novel drug development keeping in view their traditional uses of *S. cumini* for the treatment of diabetes and Alzheimer's disease. The extract, fractions, and isolated compound **1** have significant α-glucosidase-inhibiting and β-secretase-inhibiting potential compared to standard drugs. In this project, the chloroform fraction and compound **1** exhibited excellent α-glucosidase and β-secretase inhibitory effects which could be used as excellent pharmacophore templates for novel drug discovery/development. The docking study revealed significant interactions of friedelin against the target protein's binding site. Therefore, this compound might be an effective therapeutic candidate and might be effective against β-secretase and α-glucosidase.

**Author Contributions:** Conceptualization, A.M. and A.R.; methodology, S.B. (Sami Bawazeer).; software, A.W.; validation, H.A.H.; formal analysis, S.B. (Saud Bawazeer).; investigation, A.R.; resources, A.M.; data curation, S.B. (Saud Bawazeer).; writing—original draft preparation, H.A.H.; writing—review and editing, A.R.; visualization, S.B. (Sami Bawazeer).; supervision, A.W.; project administration, S.B. (Sami Bawazeer).; funding acquisition, A.M. All authors have read and agreed to the published version of the manuscript.

**Funding:** The authors extend their appreciation to the Deputyship for Research & Innovation, Ministry of Education, Saudi Arabia for funding this research work through the project number (QU-lF-05-04-28604). The authors give thanks to Qassim University for technical support.

**Data Availability Statement:** The data associated with this paper are given in the main text of this paper.

**Acknowledgments:** This work was supported by the Deputyship for Research & Innovation, Ministry of Education, and Qassim University, Saudi Arabia [QU-lF-05-04-28604]. The authors also thank Qassim University for technical support.

**Conflicts of Interest:** The authors declare no conflict of interest.

**Sample Availability:** Samples of the compounds are available from the corresponding author.

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
