# Peer review of "Isolation, Structural Elucidation, In Vitro Anti-α-Glucosidase, Anti-β-Secretase, and In Silico Studies of Bioactive Compound Isolated from Syzygium cumini L."

_processes, doi:10.3390/pr11030880_

Round 1
Reviewer 1 Report
The manuscript has serious drawbacks regardind the quality and relevance of scientific information presented. Introduction is written in a telegraphic style and needs substantial improvements for acheiving the scientific rigors and the proper form to relevate the context, aims and scope of the study.
The scientific sound of the obtained results is low, as long as chemical characterization of Friedelin was already published and its pharmacological properties were already proven.
The methodology has also major drawbacks, especially in sections 4.3, and 4.4. (how it is possible to work with that volumes in a 96-well plate???).
Unfortunately, I recommend the rejection of the manuscript because it desn't requires the minimal quality criterias for publication.
Author Response
Reviewer 1
Comments and Suggestions for Authors
The manuscript has serious drawbacks regardind the quality and relevance of scientific information presented. Introduction is written in a telegraphic style and needs substantial improvements for acheiving the scientific rigors and the proper form to relevate the context, aims and scope of the study.
Reply: Dear Reviewer, Thank you for reviewing our paper. We appreciate your valuable suggestions, which will greatly improve the quality of this manuscript.
We have carefully and comprehensively stated the aim and objective of this work in the revised manuscript, which we believe will meet the standards of rigor and clarity you require.
Thank you for taking the time to provide your feedback, and we look forward to your continued support in our efforts to improve the quality of this research.
The scientific sound of the obtained results is low, as long as chemical characterization of Friedelin was already published and its pharmacological properties were already proven.
Reply: Friedelin is a previously isolated bioactive natural product our group recently re-isolated the compound from Syzygium cumini and compare the physical and spectroscopic data with the previous one . The bioactive Friedelin is responsible for the α-Glucosidase, β-Secretase of the title plant.
The methodology has also major drawbacks, especially in sections 4.3, and 4.4. (how it is possible to work with that volumes in a 96-well plate???).
Reply: The reaction mixture was hatched with intestinal α-glucosidase for five minutes with a 50 µL substrate (5 mM, p-nitrophenyl-a-D-glucopyranoside) as per the reported method. The substrate was mixed and the difference in absorbance at 400 nm was recorded by using a spectrophotometer. The screen extract/fractions and isolated compound 1 were substituted with DMSO (7.5%) in the control. The standard used in this assay was acarbose. It is micro liter not milliliter. Working on 96 well plates is a standard procedure and normally the researchers follow/use this plate for easy and speedy work to save time. The typographical mistake has been removed now.
Unfortunately, I recommend the rejection of the manuscript because it desn't requires the minimal quality criterias for publication.
Reply: Dear sir, we have revised the paper and incorporated all needle corrections. Our paper documented the the α-Glucosidase, β-Secretase of the extract/fractions and isolated compound for the first time which is important for new drug discovery.

Reviewer 2 Report
α-Glucosidase, β-Secretase Friedelin mentioned in capital in middle of the sentences which is not right, Extraction results do not mention percentage yield of the compound (friedelin) isolated, it is one of the common triterpenoid.
Author Response
Reviewer 2
Comments and Suggestions for Authors
α-Glucosidase, β-Secretase Friedelin mentioned in capital in middle of the sentences which is not right,
Reply: Dear reviewer thank you for pointing out this. This was a typographic mistakes and has been resolved in the revised manuscript
Extraction results do not mention percentage yield of the compound (friedelin) isolated, it is one of the common triterpenoid.
Reply: Dear reviewer, the compound was purified through recrystallization in a mixture of hexane and acetone, resulting in a 99.94% pure compound of friedelin, as stated in section 4.2.

Reviewer 3 Report
Dear authors, I think you need to check very well your manuscript, regarding lenguage, spectroscopy characterization, references and in general all your discussion and conclusions.
Supplementary material will be also very useful.
Author Response
Reviewer 3
Comments and Suggestions for Authors
Dear authors, I think you need to check very well your manuscript, regarding lenguage, spectroscopy characterization, references and in general all your discussion and conclusions.
Reply: Dear Reviewer thanks aloot for your suggestion, the entire paper has been revised all the language mistake has been rectified, and the above-mentioned section (spectroscopy characterization, references, and in general all your discussion and conclusions) has been checked and improved.
Supplementary material will be also very useful.
Reply: All the figures and tables related to this paper is cited in the mean text.

Reviewer 4 Report
Comments and suggestions for authors:
This research supplied some useful and new scientific information about the extracts of Syzygium cumini L and the the isolated compound Friedelin. This work may be further reviewed and acceptable for publication after significant revision and address some major points:
1. The introduction still not provides sufficient background: In this section, the authors focused on presenting almost about the materials (medicinal plant: Syzygium cumini L and the isolated compound). How about the medical effect (Anti diabetes and anti-Alzheimer and some targeting enzymes for these diseases).
2. The logical of this work:
- Why the author chose α-glucosidase, and β-secretase for potential targeting Anti diabetes and Alzheimer, respectively, not other enzymes:
As I know, for Anti diabetes, some protein enzymes maybe mentioned: a-amylase, a-glucosidase, tyrosine phosphatase 1B (PTP1B),…
For anti-Alzheimer some protein enzymes maybe mentioned: acetylcholinesterase, butyrylcholinesterase, α-secretase, γ-secretase, glycogen synthase kinase 3 (GSK3),...
- Many parts used of this medicinal plants, why the bark was chosen for this study?
- Why only one compound was isolated? Is Friedelin a major compound of Syzygium cumini L?, or this compound was isolated based on bioassay guided purification? This need a clearer explanation in the discussion section.
- In the docking study, why no use the positive inhibitors such as acarbose and berberine chlorid for comparision.
3. Minor comments:
- Please carefully check the values of activities line 26-28:
Line 26, the IC50 of compound 1 should be17.54 μM not 290.43 µM.
Line 27-28, you mentioned “various extracts and the compound 1”, it means there are more than 3 samples, why only two values were presented. Please check.
- Under table 2 and 3, please add some information about the concentration such as: % inhibition was obtained at which concentration of samples? And what is standard compound was used?.
- In virtual study, which virtual pH was used? Please add this in the method.
Author Response
Reviewer 4
Comments and suggestions for authors:
This research supplied some useful and new scientific information about the extracts of Syzygium cumini L and the the isolated compound Friedelin. This work may be further reviewed and acceptable for publication after significant revision and address some major points:
- The introductionstill not provides sufficient background: In this section, the authors focused on presenting almost about the materials (medicinal plant: Syzygium cumini L and the isolated compound). How about the medical effect (Anti diabetes and anti-Alzheimer and some targeting enzymes for these diseases).
Reply: Dear reviewer thank you for providing your valuable comments.
We made suggested correction by adding an overview of the medicinal application of Syzygium cumini L and the isolated compound in terms of Anti diabetes and anti-Alzheimer in the introduction part.
- The logical of this work:
- Why the author chose α-glucosidase, and β-secretase for potential targeting Anti diabetes and Alzheimer, respectively, not other enzymes
Reply: α-glucosidase and β-secretase are specific enzymes targeted in the treatment of diabetes and Alzheimer's disease, respectively, because of their unique biological activities.
Choosing these specific enzymes for potential targeting based on their specific biological activities in diabetes and Alzheimer's disease, respectively, allows for a more targeted and effective approach to treatment, compared to targeting other enzymes that may have less direct involvement in the pathogenesis of these conditions.
As I know, for Anti diabetes, some protein enzymes maybe mentioned: a-amylase, a-glucosidase, tyrosine phosphatase 1B (PTP1B),…
For anti-Alzheimer some protein enzymes maybe mentioned: acetylcholinesterase, butyrylcholinesterase, α-secretase, γ-secretase, glycogen synthase kinase 3 (GSK3),...
Reply: Based on the folkloric usage of the plant we have selected it for α-Glucosidase, β-Secretase inhibitory activity. Also the main aim of this project was to identify the bioactive compound -responsible for Glucosidase, β-Secretase. In the future project, we agree with the referees’ comments we will do the extensive screening.
- Many parts used of this medicinal plants, why the bark was chosen for this study?
Reply: The bark of Syzygium cumini L is a better choice for the isolation of biologically active compounds as compared to the roots and stems, due to its higher concentration of bioactive compounds, availability, cost-effectiveness, traditional use, high antioxidant activity, ease of extraction, and standardized protocols.
- Why only one compound was isolated? Is Friedelin a major compound of Syzygium cumini L?, or this compound was isolated based on bioassay guided purification? This need a clearer explanation in the discussion section.
Reply: In the docking study, why no use the positive inhibitors such as acarbose and berberine chlorid for comparision. Thanks for your suggestions the docking analysis of Acarbose and berberine chloride is now added.
Minor comments:
- Please carefully check the values of activities line 26-28:
Reply: Corrected
Line 26, the IC50 of compound 1 should be17.54 μM not 290.43 µM.
Reply: Thank you for pointing out this. The corrections have been made in the thought the manuscript.
Line 27-28, you mentioned “various extracts and the compound 1”, it means there are more than 3 samples, why only two values were presented. Please check.
Reply: The crude extract, fractions and compound 1 were used in this study. It was typo error corrected now.
- Under table 2 and 3, please add some information about the concentration such as: % inhibition was obtained at which concentration of samples? And what is standard compound was used?.
Reply: Concentrations (µg/ml) % Inhibition IC50±SEM (µM). it is already mentioned in the table heading
- In virtual study, which virtual pH was used? Please add this in the method.
Reply: In virtual the default neutral pH was used.

Round 2
Reviewer 1 Report
I saw that the authors have introduced several changes in the content of manuscript in order to improve its quality. However, the introduction still have important drawbacks, especially lines 39-56 (the information has no relevance to the present study and text is written in a telegraphic style).
Lines 77-86: aims and scope must be presented using past. Please, check and revise accordingly.
Lines 140, 142, 152: check and revise the title of figures and tables; add meaning of SEM from the head of tables.
Sections 4.1. and 4.2: The plant material was freshly extracted or previously processed (dried, etc.); please mention this aspect. As well, please clarify the meaning of „cold extraction” (the method used - maceration, percolation - and parameters - temperature).
Discussion section is focused mainly on general facts regardind diabetes mellitus and Alzheimer disease, instead of on the importance of your obtained results in the context of this pathologies. Moreover, try to compare your obtained resulth with those provided by other studies regarding the same species or similar species in order to emphaside the importance and relevance of your obtained results.
Conclusion section: Please , replace „pharmacophore templates” with an appropriate scientific term (i.e. scaffold, lead compound - feel free to use other than those suggested, in order to keep a proper scientific meaning).
Lines 277-277: please, rephrase; there is no link and meaning between the two phrases.
As a general recommendation, please, revise English drawbacks from the whole manuscript.
Author Response
Dear Editor, Thank you for your comments. We have addressed all the comments and highlighted them in the manuscript.
Reviewer 1
Comments and Suggestions for Authors
I saw that the authors have introduced several changes in the content of manuscript in order to improve its quality. However, the introduction still have important drawbacks, especially lines 39-56 (the information has no relevance to the present study and text is written in a telegraphic style).
Reply: Dear reviewer, thank you again for reviewing our paper.
According to the suggestion the mentioned text in the introduction has been revised accordingly
Lines 77-86: aims and scope must be presented using past. Please, check and revise accordingly.
Reply: Dear reviewer, thank you for your suggestion. The suggested corrections have been made in the revised manuscript and are highlighted
Lines 140, 142, 152: check and revise the title of figures and tables; add meaning of SEM from the head of tables.
Reply: Dear reviewer, thank you for pointing out this. The suggested corrections have been made in the revised manuscript and is highlighted.
Sections 4.1. and 4.2: The plant material was freshly extracted or previously processed (dried, etc.); please mention this aspect. As well, please clarify the meaning of „cold extraction” (the method used - maceration, percolation - and parameters - temperature).
Reply: The dried plant material was subjected to cold extraction via maceration, using 70% methanol as the solvent, at room temperature or slightly lower for a duration of two weeks. The suggested correction has been made by revising the 1st paragraph of the section 4.2 and is highlighted.
Discussion section is focused mainly on general facts regardind diabetes mellitus and Alzheimer disease, instead of on the importance of your obtained results in the context of this pathologies. Moreover, try to compare your obtained resulth with those provided by other studies regarding the same species or similar species in order to emphaside the importance and relevance of your obtained results.
Reply: The needful corrections have been done in the revised paper. Thank you. The results have been mentioned there is no previous data on this species extract or compounds. We have explored this for the first time first.
Conclusion section: Please , replace „pharmacophore templates” with an appropriate scientific term (i.e. scaffold, lead compound - feel free to use other than those suggested, in order to keep a proper scientific meaning).
Reply: Dear reviewer, thank you for your valuable suggestion, the word “pharmacophore templates” has been replaced by “Molecular scaffold” in the revised manuscript and is highlighted
Lines 277-277: please, rephrase; there is no link and meaning between the two phrases.
Reply: The mentioned sentence has been revised in the revised manuscript.
As a general recommendation, please, revise English drawbacks from the whole manuscript.
Reply: The English have been edited throughout the text.

Reviewer 2 Report
The article shall be accepted.
Author Response
Reviewer 2
Comments and Suggestions for Authors
The article shall be accepted.
Reply: Thanks aloot.

Reviewer 3 Report
The manuscript need improve, some isuues like the spectroscopy description, for example:
is not triplite is triplet
is not multiplate is multiplet
Is not posible a triplet with two differents J, also if you describe and quartet or quintuplet, you need give the J values.
In general the spectroscopic description need to be improved.
You have in the references 36 and in the text only 32 are mentioned.
Author Response
Reviewer 3
Comments and Suggestions for Authors
The manuscript need improve, some isuues like the spectroscopy description, for example:
is not triplite is triplet.
Reply: Corrected as suggested
is not multiplate is multiplet.
Reply: Corrected as suggested
Is not posible a triplet with two differents J, also if you describe and quartet or quintuplet, you need give the J values.
Reply: We would like to thank the Reviewer for the kind suggestions for improving the manuscript. All corrections have been included in the revised manuscript. It was a typo mistake the proton at C-8 appeared as a doublet of doublets (dd) at 1.38.
In general the spectroscopic description need to be improved.
Reply: The spectroscopic description has been improved.
You have in the references 36 and in the text only 32 are mentioned.
Reply: Corrected throughout the text.

Reviewer 4 Report
Almost the comments were addressed and answered.
I suggested "accepeted" for publiaction with minor modification of the tittle for exact meaning:
Suggested tittle" Isolation, Structure Elucidation, in vitro anti α-glucosidase, anti β-secre- 2 tase, and in silico studies of Bioactive Compound isolated from 3 Syzygium cumini L"
At [line 27], please revise "compoun1" as "compound 1"
Author Response
Reviewer 4
Comments and Suggestions for Authors
Almost the comments were addressed and answered.
I suggested "accepeted" for publiaction with minor modification of the tittle for exact meaning:
Suggested tittle" Isolation, Structure Elucidation, in vitro anti α-glucosidase, anti β-secre- 2 tase, and in silico studies of Bioactive Compound isolated from 3 Syzygium cumini L"
Reply: Corrected as suggested.
At [line 27], please revise "compoun1" as "compound 1"
Reply: Corrected as suggested.

Round 3
Reviewer 1 Report
In my opinion, the manuscript content has been considerable improved so it could be published in the present form, after minor changes. Please, check again the text for english spelling and errors.
Lines 55-56, replace 'utilized' with 'used', 'make stronger of the teeth and the gums' with 'to improve teeth and gums health';
Lines 57-58: delete 'Syzygium cumini L, also known as Indian black plum or Jamun, has 57 been traditionally used for various medicinal purposes' and include the antidiabetic activity to the above-mentioned enumeration and make one sentence.
Lines 62-63: 'The plant has been found to 62 reduce fasting blood sugar levels, improve insulin sensitivity, and lower oxidative stress 63 in diabetic individuals' please provide some reference(s) that can adequately support this statement (prefferable data from in vivo or clinical studies).
Lines 86-87: 'Compound 1 namely friedelin was isolated from chloroform soluble fractions of S. cumini stem'. It is stem or stem's bark? In materials and method section you have specified that plant material was represented by barks... Please, check and revise accordingly in the content of the manuscript this error.
Author Response
Reviewer 1
Comments and Suggestions for Authors
In my opinion, the manuscript content has been considerable improved so it could be published in the present form, after minor changes. Please, check again the text for english spelling and errors.
Reply: Thank you.
Lines 55-56, replace 'utilized' with 'used', 'make stronger of the teeth and the gums' with 'to improve teeth and gums health';
Reply: Corrected as suggested.
Lines 57-58: delete 'Syzygium cumini L, also known as Indian black plum or Jamun, has 57 been traditionally used for various medicinal purposes' and include the antidiabetic activity to the above-mentioned enumeration and make one sentence.
Reply: Corrected as suggested.
Lines 62-63: 'The plant has been found to 62 reduce fasting blood sugar levels, improve insulin sensitivity, and lower oxidative stress 63 in diabetic individuals' please provide some reference(s) that can adequately support this statement (prefferable data from in vivo or clinical studies).
Reply: Corrected as suggested. Reference no 17 has been added now. (Hussein, J.; El-Khayat, Z.; Taha, M.; Morsy, S.; Drees, E.; Khateeb, S. Insulin resistance and oxidative stress in diabetic rats treated with flaxseed oil. J Med Plants Res 2012, 6(42), 5499-5506).
Lines 86-87: 'Compound 1 namely friedelin was isolated from chloroform soluble fractions of S. cumini stem'. It is stem or stem's bark? In materials and method section you have specified that plant material was represented by barks... Please, check and revise accordingly in the content of the manuscript this error.
Reply: The compound is isolated from the bark. Corrected throughout text now. Thank you

Reviewer 3 Report
Dear authors
The review of this manuscript in the NMR spectroscopic description, has not yet been corrected. The errors that I previously pointed out to you continue, it is not only the words that should be corrected.
Multiplicity not means CH2, CH3, C, it means the signal form: doublet, triplet, etc.
For example, if in NMR a signal is described with its multiplicity, it is like a doublet, triplet, quartet, doublet of doublet, the values of the J constant must be specified, this is an important data which gives very important information of the structure of the anlyzed compound.
I want to express that I have no doubts in the structure of compound 1, its is Friedelin, according to the spectroscopy data showed in the manuscript, however my insistence is that I believe that the way in which the NMR data are presented should be corrected, the help of chemist colleague would be useful.
Author Response
Reviewer 3
Comments and Suggestions for Authors
Dear authors
The review of this manuscript in the NMR spectroscopic description, has not yet been corrected. The errors that I previously pointed out to you continue, it is not only the words that should be corrected.
Multiplicity not means CH2, CH3, C, it means the signal form: doublet, triplet, etc.
For example, if in NMR a signal is described with its multiplicity, it is like a doublet, triplet, quartet, doublet of doublet, the values of the J constant must be specified, this is an important data which gives very important information of the structure of the anlyzed compound.
I want to express that I have no doubts in the structure of compound 1, its is Friedelin, according to the spectroscopy data shown in the manuscript, however, my insistence is that I believe that the way in which the NMR data are presented should be corrected, the help of chemist colleague would be useful.
Reply: We would like to thank the Reviewer for the kind suggestions for improving the manuscript. The NMR data has been checked once again and corrected. This is a known compound the chemical structure of this compound is identified by comparing the spectra data with report one.
